# Fatty Pancreas: Its Potential as a Risk Factor for Pancreatic Cancer and Clinical Implications

**DOI:** 10.3390/cancers17111765

**Published:** 2025-05-24

**Authors:** Nao Otsuka, Yutaka Shimamatsu, Ryunosuke Hakuta, Yukiko Takayama, Yousuke Nakai

**Affiliations:** Department of Internal Medicine, Institute of Gastroenterology, Tokyo Women’s Medical University, Tokyo 162-8666, Japan; okuno.nao@twmu.ac.jp (N.O.); shimamatsu.yutaka@twmu.ac.jp (Y.S.); hakuta-tky@umin.ac.jp (R.H.); takayama.ige@twmu.ac.jp (Y.T.)

**Keywords:** fatty pancreas, intra-pancreatic fat deposition (IPFD), pancreatic ductal adenocarcinoma (PDAC)

## Abstract

Fatty pancreas, a condition in which fat builds up in the pancreas, is increasingly detected with modern imaging techniques. While its impact remains unclear, it is linked to obesity and diabetes and may increase the risk of pancreatic cancer. Understanding this connection could help identify high-risk individuals and improve early detection of pancreatic cancer. This study explores the relationship between fatty pancreas, metabolic disorders, and pancreatic cancer, highlighting potential prevention and treatment strategies.

## 1. Introduction

The increase in utilizing imaging modalities such as transabdominal ultrasonography (US), computed tomography (CT), and magnetic resonance imaging (MRI) has led to the diagnosis of more incidental pancreatic abnormalities, especially pancreatic cysts. Fatty change in the pancreas is also encountered on those cross-sectional images, but its clinical significance has not yet been clarified. Fatty pancreas (FP), which is also referred to as pancreatic steatosis (PS), intra-pancreatic fat deposition (IPFD), or fatty pancreas disease (FPD), is defined as an accumulation of fat in the pancreas. FP has recently attracted attention in relation to metabolic syndromes such as obesity and diabetes [1,2,3]. Interestingly, those metabolic factors are associated with pancreatic cancer as well as pancreatic cystic neoplasms [4], including intraductal papillary mucinous neoplasms (IPMNs), which are known precursors of pancreatic ductal adenocarcinoma (PDAC). Although metabolic dysfunction-associated steatotic liver disease (MASLD) and its hepatic consequences, such as cirrhosis and hepatocellular carcinoma, have been extensively studied, FP has only recently emerged as a focus of investigation, potentially representing a pancreatic counterpart to MASLD. There is increasing evidence that FP is associated with pancreatic carcinogenesis [5,6,7,8]. Although early detection of PDAC is still difficult, it can lead to better survival, suggesting the importance of surveillance in high-risk individuals, such as those with new-onset diabetes [9], pancreatic cysts [10], or a familial or genetic predisposition for PDAC [11]. This review highlights the current knowledge of FP, its underlying mechanisms, its relation to obesity or metabolic syndromes, its role as a risk factor for PDAC development, and potential treatment options.

## 2. Pathology

In the human pancreas, fatty infiltration is observed both in intralobular and extralobular (peri-lobular) locations (Figure 1). In the normal pancreatic tissue from the surgical specimens who underwent surgical resection of benign and small pancreatic neuroendocrine neoplasms, fatty infiltration was found in 51% in intralobular and 30% in extralobular locations [12].

Intralobular fat means lipid droplets in acinar cells, lipid droplets in the islets of Langerhans, acinar-to-adipocyte trans-differentiation, replacement of apoptotic acinar cells, or a combination thereof. IPFD is morphologically characterized by the presence of fat-containing cells between lobules (interlobular fat or peri-lobular fat) and intralobular fat [13].

Frendi et al. [8] defined the extralobular fat (ELF) infiltration as the presence of adipocytes outside the acinar lobules, mainly located in the interlobular and peri-lobular space, and evaluated the lipidomic profiles of ELF and pancreatic intralobular fat (ILF). This study showed that the lipid composition of ELF and ILF was different, and ELF-specific lipids were linked to the development of pancreatic intraepithelial neoplasia (PanIN), while ILF was associated with obesity [8]. Additionally, acinar cells exhibited distinct phenotypes based on the presence and proximity of ILF in obese patients.

In our previous study [14] on the pathological analysis of FP, substantial fatty infiltration into the acinar regions originating from the pancreatic interstitium was observed, along with atrophy of the pancreatic parenchyma. Both M1 and M2 macrophages were identified within the areas of fatty infiltration [14]. Whether the atrophy caused by fatty infiltration or the presence of these macrophages contributes to inflammation, fibrosis, or carcinogenesis remains an issue for future research.

## 3. Physiopathology and Risk Factors

As suggested above, IPFD occurs through two mechanisms: (1) “fatty replacement” through acinar cell death and adipocyte replacement, or (2) “fatty infiltration” or accumulation associated with obesity and/or metabolic syndrome [15].

“Fatty replacement” is the death of pancreatic acinar cells followed by their replacement with adipocytes. This process can be caused by alcohol abuse; possibly chronic liver disease—including MASLD, viral hepatitis, and cirrhosis—as suggested in some reports [16,17]; congenital diseases (such as cystic fibrosis, Shwachman–Bodian–Diamond syndrome, and Johanson–Blizzard syndrome); iron overload; malnutrition; medications (corticosteroids, gemcitabine, octreotide, and rosiglitazone); pancreatitis (acute and chronic); and viral infections with reovirus [18,19].

Fatty replacement is considered irreversible. In animal studies, pancreatic duct ligation caused increased pancreas volume due to edema in the first two days, followed by acinar cell apoptosis. After two weeks, fatty replacement began, restoring pancreas volume within eight weeks [20]. Several human observational studies showed that pancreatic damage leading to acinar cell necrosis resulted in fatty replacement. Recurrent acute pancreatitis (AP) could decrease the parenchymal mass and replace it with adipocytes [21,22].

“Fatty infiltration” is the accumulation of adipocytes within pancreatic tissue, primarily associated with obesity [23,24]. Adipose tissue functions as an endocrine organ, and excessive fat storage can lead to ectopic fat deposition in non-adipose tissues such as the liver, skeletal muscle, and pancreas [25]. Sedentary lifestyles with low exercise, obesogenic diets, metabolic syndrome, and differential storage of visceral fat increase the risk of ectopic fat deposition in the organs [26]. Pancreatic hypertrophy and hyperplasia occur in the early stages of obesity, and these changes are followed by fat infiltration of the pancreas [27]. The accumulated fat may contribute to insulin resistance, β-cell dysfunction, and the subsequent development of type 2 diabetes mellitus (T2DM) [28]. This progression is thought to be mediated by prolonged exposure to elevated fatty acids, which impairs insulin secretion and disrupts gene expression in the islets of Langerhans [29,30]. Fatty infiltration is considered potentially reversible [31]. The fact that several drugs have been shown to reduce pancreatic fat content [32] provides evidence of its reversibility.

## 4. Prevalence

The definition and diagnostic criteria for FP have not been established, which makes the evaluation of its prevalence difficult. A systematic review and meta-analysis including 11,967 individuals in 17 studies [33] revealed that the prevalence of FP was 12.9% to 30.7% in healthy individuals. In these studies, the prevalence differs even though the same imaging modality is used to assess fatty infiltration of the pancreas (Table 1). As will be described later in the Diagnosis section, the threshold for diagnosing FP using CT attenuation values and the fat fraction of normal pancreas by MRI is not well-defined. Therefore, the prevalence assessed by CT is unknown, and only one study by Wong et al. has shown a prevalence of 16.1% using MRI [34]. FP was associated with metabolic factors such as triglycerides (TG) and insulin resistance. Given the global increase in metabolic syndrome, fatty change of the pancreas would further increase in the future.

## 5. Diagnosis

Given the lack of standard diagnostic criteria, the inter-study agreement of the diagnostic yield is difficult, and comparison among the diagnostic modalities is impossible. Various diagnostic criteria for each imaging modality are used as follows.

### 5.1. Transabdominal Ultrasonography (US)

Wang et al. [35] reported FP was diagnosed when there was an increase in echogenicity of the pancreatic body over that of the kidney. Because the pancreas often could not be directly compared with the kidney within the same imaging window (Figure 2), the examiner assessed the differences between hepatic and renal echogenicity, and between hepatic and pancreatic echogenicity, to obtain an objective pancreatorenal echo contrast [35]. In another study, pancreatic echogenicity was compared to the liver echogenicity at the same depth on a longitudinal scan taken near the abdominal midline [36]. Figure 2 shows ultrasonographic images of a non-fatty pancreas and a fatty pancreas.

### 5.2. Endoscopic Ultrasound (EUS)

Sepe et al. [23] reported that the evaluation of FP by EUS was based on comparing the echogenicity of the pancreas with that of the spleen, as well as assessing the clarity of the main pancreatic duct contour. If the pancreas was hypo/isoechoic or hyperechoic and the main pancreatic duct margin was clearly delineated, it was considered normal, and if it was moderately to severely hyperechoic and the main pancreatic duct margin was moderately/severely obscured, it was considered FP [23]. Figure 3 shows EUS images of a non-fatty pancreas and a fatty pancreas.

### 5.3. Computed Tomography (CT)

CT attenuation values can be measured in Hounsfield units (HU) (Figure 4). Kim et al. [38] reported that CT attenuation values can be used for a relatively good semiquantitative estimation of pancreatic fat content, as their findings demonstrated a correlation between CT parameters and histologically confirmed fat fraction. The decrease in pancreatic CT attenuation values on non-contrast enhanced CT images correlates with the degree of pancreatic fat infiltration [38,39]. The pancreas-to-spleen (P/S) CT attenuation value ratio is used to assess pancreatic fat. In addition, the absolute difference in CT attenuation values between the pancreas and spleen (P–S) has also been used. Additionally, Mori et al. [39] reported that when the P/S CT attenuation value ratio is less than 0.8, moderate or higher fatty deposits are observed in the pancreatic parenchyma in the histological evaluation of FP. This CT attenuation can be simply and easily measured in clinical practice.

### 5.4. Magnetic Resonance Imaging (MRI)

MRI, especially MR spectroscopy (MRS), enables the non-invasive in vivo quantification of tissue fat content [40,41]. However, MRS is time-consuming, limited to assessing fat content in specific organ areas, and requires expert voxel placement, particularly for small or irregularly shaped organs. Its accuracy may also be affected by organ shifts caused by diaphragm motion during breathing [41,42]. Recently, in clinical studies of fatty liver, the MR Dixon technique has emerged as another widely used method for fat quantification, offering greater accuracy than MRS or liver biopsy in some studies [42,43]. It leverages the in-phase/out-of-phase cycling of fat and water to create fat-only and water-only images, making it easier to perform than MRS [41]. Fat and water images can be obtained within a single breath-hold, reducing motion-related errors. Additionally, it provides a comprehensive view of fat distribution across internal organs, allowing whole-liver assessment rather than only a specific region of interest (ROI) [44]. While extensively used for liver fat analysis, its application to pancreatic fat quantification remains limited. Notably, MRI scan protocols for the pancreas vary between studies [34,44,45,46,47], and different methods of analyzing MR data result in measurement variability of pancreatic fat content. A systematic review recommended 6% as the threshold for FP [2], while another study suggested that an upper limit of normal for pancreatic fat is 10.4% [34].

While both CT and MRI are used for evaluation of the visceral fat, MRI is becoming more popular because it does not involve radiation exposure and allows for accurate quantification. However, the routine use of MRI is not practical due to its high cost, time-consuming nature, and complex protocols. CT, which allows for quick quantification, even of a retrospective nature, is more convenient in clinical practice, aside from the issue of radiation exposure.

Pancreatic fat measurement is not currently recommended for routine screening, but it may be considered in selected high-risk individuals. CT and MRI are the most commonly used modalities for quantitative or semiquantitative assessment in research and, increasingly, in clinical settings.

### 5.5. Biomarkers

Several studies have reported a correlation between biomarkers and pancreatic fat percentage, suggesting that TG and hemoglobin A1c (HbA1c) are useful markers in meta-analyses [33]. These markers will be valuable tools for the early detection of pancreatic fat excess and the reduction of FP and related metabolic diseases. Wong et al. [34] reported that hypertriglyceridemia and hyperferritinemia are associated with FP, and individuals with FP have increased insulin resistance, suggesting this population as a target for surveillance or medical interventions.

## 6. Clinical Characteristics

The involvement of FP in metabolic syndrome-related diseases such as obesity, diabetes, hypertension, and fatty liver has become increasingly evident. The presence of FP was significantly higher in central obesity (odds ratio [OR] 5.755, 95% confidence interval [CI] 3.748–8.837, *p* < 0.01), diabetes (OR 1.521, 95%CI 1.084–2.135, *p* < 0.05), hypertriglyceridemia (OR 1.346, 95%CI 1.008–1.797, *p* < 0.05), and hepatic steatosis (OR 2.522, 95%CI 1.830–3.475, *p* < 0.01) [37], and it was reported that FP was associated with a 67% higher risk of hypertension and a 108% higher risk of diabetes mellitus [2]. FP and β-cell function are suggested to be negatively associated with each other in non-diabetic individuals [48] as well as in those with impaired fasting glucose or impaired glucose tolerance [49].

Emerging evidence also indicates that excess body fat is associated with ectopic fat accumulation in the liver and pancreas [33]. There are many studies on the relationship with fatty liver, particularly nonalcoholic fatty liver disease (NAFLD) or MASLD [1,35,36,37,50,51]. FP is observed as a comorbidity in 37–80% of patients with NAFLD or MASLD [17,51,52]. In fact, MRI-estimated pancreatic fat content was significantly greater in patients with NAFLD than in healthy controls (8.5% vs. 3.6%, *p* < 0.001), and the fat content of the liver was significantly correlated with that of the pancreas (r = 0.57, *p* < 0.001) [53]. In addition, the strong correlation between the fat content of the pancreas and liver suggests that, regardless of the presence of NAFLD, pancreatic fat may be a marker for ectopic fat deposition in other organs [53]. However, the severity or activity grade of NAFLD has not been shown to be associated with progressive fat infiltration in the pancreas [54]. Moreover, whether fatty infiltration of the pancreas causes inflammation or fibrosis and leads to pancreatic dysfunction is less well established than it is for liver damage in NAFLD or MASLD [51].

Regarding exocrine function, FP is suggested to contribute to the severity of AP. As a background, it is believed that excessive adiposity leads to an increased amount of intrapancreatic fat, which, in turn, causes acute lipotoxicity, resulting in pancreatic acinar cell necrosis and systemic inflammation during AP [55]. In addition, new-onset diabetes after AP is also being actively investigated. IPFD and visceral fat volume are associated with the presence of diabetes after AP [45]. It is reported that individuals with diabetes after AP had significantly higher intrapancreatic fat percentage and visceral fat volume compared with individuals without diabetes after AP and healthy controls.

Endoscopic retrograde cholangiopancreatography (ERCP) is often utilized for the management of pancreatobiliary diseases, but post-ERCP pancreatitis (PEP) is an unsolved problem. FP is also a risk factor for PEP [56,57]. Park et al. [56] reported that FP was significantly associated with the development of PEP (OR 2.38, 95% CI 1.16–4.87). Furthermore, it was reported that the risk for moderate-to-severe PEP development tended to be higher in the FP group than in the no-FP group (OR 5.61, 95%CI 0.63–49.62). Prouvot et al. [57] also reported that FP was statistically associated with PEP (OR 7.94, 95% CI 1.59–31.09, *p* = 0.005), even after adjustment for age and sex. Therefore, clinicians need to be aware of FP as a risk factor for PEP, and prophylaxis for PEP should be considered in cases with FP.

Recently, the association of FP with chronic pancreatitis (CP) has also garnered increasing attention. Tirkes et al. [46] have reported that patients with chronic pancreatitis (CP) had a higher pancreatic fat content by pancreatic fat fraction: 21% in the severe CP group (*p* = 0.02), 23% in moderate CP (*p* < 0.0001), 24% in mild CP (*p* < 0.0001), and 15% in the no-CP group [46].

## 7. Carcinogenesis

FP is also associated with intraductal papillary mucinous neoplasm (IPMN), precancerous lesions, and pancreatic ductal adenocarcinoma (PDAC).

Among the animal models that recapitulate human PDAC, pancreas-specific activation of oncogenic *K-ras* exhibits the full spectrum of PanIN lesions, closely resembling the human malignancy from tumor initiation to progression [58]. Recent studies suggest that, in addition to somatic *K-ras* activation, the development of PDAC requires non-genetic factors such as increased inflammation and tissue damage [59]. IPFD may play a role in promoting inflammation, as adipose infiltration involves TG and free fatty acids, which contribute to fibrogenesis through necrosis and the direct activation of pro-inflammatory pathways. This inflammatory microenvironment is thought to contribute to the initiation of precancerous pancreatic lesions [60]. Consistently, IPFD has been associated with increased pancreatic expression of pro-inflammatory and fibrogenic markers, including interleukin-6 (IL-6), tumor necrosis factor-α (TNF-α), transforming growth factor-β (TGF-β), α-smooth muscle actin (α-SMA), and monocyte chemoattractant protein-1 (MCP-1) [15,61,62]. FP thus may increase the risk of the development and progression of PDAC through the secretion of adipokines that stimulate inflammation, antagonize apoptosis, and promote cell proliferation and migration. Based on these reports, the potential impact of intra-pancreatic fat deposition (IPFD) on pancreatic carcinogenesis is illustrated in Figure 5.

To determine whether the pro-inflammatory environment caused by FP leads to any pathological changes, our group previously investigated the pathophysiological mechanisms of pancreatic cancer by analyzing the factors associated with fatty infiltration of the pancreas [14]. It revealed that the accumulation of p62, related to autophagy dysfunction, and NAD(P)H quinone oxidoreductase 1 (NQO1), an antioxidant defense enzyme that protects cells from oxidative stress by neutralizing harmful free radicals, were observed in pancreatic acinar cells, acinar-ductal metaplasia, and pancreatic cancer cells within the fatty area [14]. Furthermore, the rate of p62 showed strong positive correlations with the rate of fat fraction (between p62 positive area and fatty area; r = 0.6178, *p* = 0.0037). Similarly, NQO1 exhibited the same trend (between NQO1 positive area and fatty area; r = 0.7202, *p* = 0.0003) [14]. Further research is needed to determine whether p62 accumulation and autophagy dysfunction are involved in carcinogenesis.

In fact, pancreatic fat volume reportedly correlates with the risk of pancreatic cancer. In a case control study of pancreatic cancer resection and non-cancer pancreatic resection, the pancreatic fat volume was higher in the pancreatic cancer resection cases (median fat infiltration 26% vs. 15%, *p* < 0.001), and those with higher pancreatic fat volumes had a significantly higher OR for pancreatic cancer (fat infiltration area <10% vs. ≥20%: OR 6.1, 95% CI 2.4–15.2, *p* < 0.001) [5].

As the association between FP and pancreatic cancer becomes clearer, discussions arise as to whether FP is a secondary change caused by pancreatic cancer. Interestingly, there was no difference in the degree of pancreatic steatosis before and after PDAC diagnosis on CT scans, suggesting that FP, rather than resulting from cancer-associated inflammation, is a carcinogenic risk factor [63].

As supporting evidence of FP as a risk for pancreatic carcinogenesis, an association of FP with precancerous lesions was also reported. The presence of pancreatic cancer or pre-malignant lesions was associated with a significantly increased risk of IPFD (relative risk 2.78, 95% CI 1.56–4.94, *p* < 0.001) [7]. In particular, its association with PanIN, a pre-malignant lesion, has been increasingly recognized. It has become evident that ILF contributes to the development of PanIN. Moreover, pathologically, the fact that pancreatic fat infiltration is not limited to the area surrounding PanIN suggests that PanIN may arise from FP [12]. Frendi et al. [8] have shown that the number of PanINs was higher in obese patients and revealed that there are two types of pancreatic fat infiltration, ILF and ELF, with different lipid compositions that play different roles in the oncogenesis process, especially in obese patients. It is suggested that ILF plays a major role in acinar modifications and the development of precancerous lesions associated with obesity, while ELF may play a role in the progression of PDAC [8].

FP is also associated with cyst enlargement in branch duct IPMN (BD-IPMN). It has been shown that pancreatic fat content and initial cyst size were significantly associated with imaging progression in low-risk BD-IPMN. CT attenuation indexes, the difference between the pancreas and spleen attenuation and the pancreas to spleen attenuation ratio, were significantly lower in the progression group than in the non-progression group (−11.1 ± 7.4 vs. −5.5 ± 6.5, *p* = 0.012; and 0.80 ± 0.10 vs. 0.87 ± 0.12, *p* = 0.048, respectively) [64]. In other words, cases with cyst enlargement have significantly more pancreatic fat content. Thus, surveillance of IPMN can be individualized by pancreatic fat content in addition to cyst size [65].

In summary, FP is associated with both the development of PanIN and cyst enlargement of BD-IPMN and can be an important factor in the pancreatic carcinogenesis process.

## 8. Interventions

Given the potential role of FP in pancreatic carcinogenesis, interventions for FP might prevent pancreatic cancer development. It has been shown that several medications may reduce pancreatic fat. Notably, some newer glucose-lowering medications, which also potentially work for MASLD, have been investigated for improvement of IPFD: glucagon-like peptide-1 (GLP-1) receptor agonists, dipeptidyl peptidase-4 (DPP-4) inhibitors, and sodium-glucose cotransporter-2 (SGLT-2) inhibitors (Table 2).

### 8.1. GLP-1 Receptor Agonists

Five clinical studies (four randomized controlled trials [RCTs] and one retrospective cohort study) investigated changes in IPFD following the administration of GLP-1 receptor agonists in individuals with T2DM [66,67,68,69,70]. Vanderheiden et al. [67] reported that liraglutide did induce significant weight loss and improved β-cell function, and liver fat and abdominal subcutaneous adipose tissue decreased in the liraglutide group vs. placebo, whereas visceral adipose tissue did not change significantly between the groups. Pancreatic fat content improved minimally in the liraglutide group and remained unchanged in the placebo group, but without statistical significance between the two groups [67], which was in line with other studies [66,68,69]. Additionally, a recent retrospective study in Japan on 13 individuals with T2DM on liraglutide over an 11-year study period found that the change in IPFD on CT before and after liraglutide administration decreased but narrowly missed statistical significance (*p* = 0.0547) [70]. It is possible that the reduction of ectopic fat from the pancreas requires a longer-term or more potent intervention compared with MASLD.

### 8.2. DPP-4 Inhibitor

One clinical study investigated changes in IPFD following the administration of a DPP-4 inhibitor [68]. This RCT of sitagliptin vs. placebo in addition to standard care for T2DM showed a significant decrease in IPFD, with an absolute reduction in IPFD of 4.2%.

### 8.3. SGLT-2 Inhibitors

Five (four prospective and one retrospective) clinical studies investigated changes in IPFD following the administration of SGLT-2 inhibitors in individuals with T2DM or prediabetes [71,72,73,74,75]. Three studies [71,74,75] showed that SGLT-2 inhibitors significantly decreased pancreatic fat content. One RCT [73], on the other hand, showed that after an 8-week treatment with empagliflozin, pancreatic fat content did not decrease significantly. This discrepancy might be due to the short study period or differences in the study population of T2DM alone.

### 8.4. Other Medications and Treatments

Other possible classes of medications for FP are statins and angiotensin II receptor blockers [76]. The existence of a local renin–angiotensin system in the pancreas was reported and might be associated with IPFD [77]. In addition to these medications, the effectiveness of exercise training and diets has also been shown. Several RCTs have shown the benefits of certain diets (low-carbohydrate diets or diets enriched with polyunsaturated fatty acids) [78,79,80,81,82] and exercise training (sprint interval training or moderate-intensity continuous training) [83]. Notably, Heiskanen [83] reported that two weeks of exercise training improved beta-cell function in individuals with prediabetes or type 2 diabetes and significantly decreased pancreatic fat, regardless of baseline glucose tolerance. This finding suggests that even short-term training can efficiently reduce ectopic fat within the pancreas and potentially lower the risk of type 2 diabetes.

Petrov [84] proposed the hypothesis of the “PANDORA model” (PANcreatic Diseases Originating from intRapancreaticfAt), which integrates pancreatic exocrine and endocrine functions. In this PANDORA hypothesis, interventions on IPFD could play a key role not only in pancreatic endocrine disorders but also in the prevention (or inhibition) of exocrine pancreatic diseases. This is particularly important because, while numerous treatments target the pathophysiology of T2DM, there are currently no interventions available to halt the natural progression of pancreatitis or sporadic pancreatic cancer [84].

## 9. Conclusions

There are no consensus guidelines yet on diagnosing or evaluating FP. As FP is expected to be an important risk factor for pancreatic cancer, its evaluation could potentially contribute to improved surveillance and early diagnosis of pancreatic cancer, particularly by identifying high-risk groups. Additionally, the development of interventions for the prevention and improvement of FP should be further explored, and whether these interventions can reduce the incidence of pancreatic cancer should be further highlighted as an important research topic going forward.

## Figures and Tables

**Figure 1 cancers-17-01765-f001:**
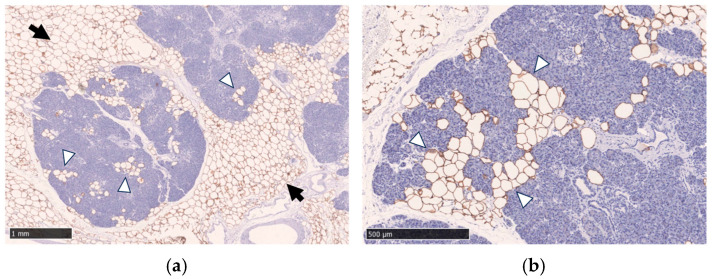
Immunohistochemical staining with anti-perilipin-1 antibody and hematoxylin counterstain demonstrates fatty infiltration in the pancreas. Fatty infiltration in the intralobular location (white arrowheads) and the extralobular location (black arrows) are shown. Immunohistochemical staining was performed using an anti-perilipin-1 antibody to highlight lipid droplets. Scale bar: 1 mm in (**a**) and 500 μm in (**b**).

**Figure 2 cancers-17-01765-f002:**
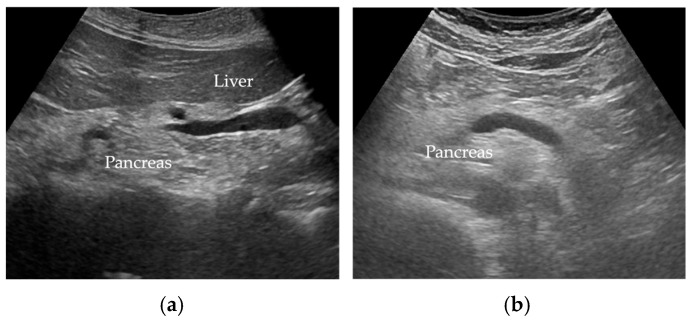
Ultrasonographic images show (**a**) a non-fatty pancreas with normal echogenicity and (**b**) a fatty pancreas with increased echogenicity of the pancreatic body.

**Figure 3 cancers-17-01765-f003:**
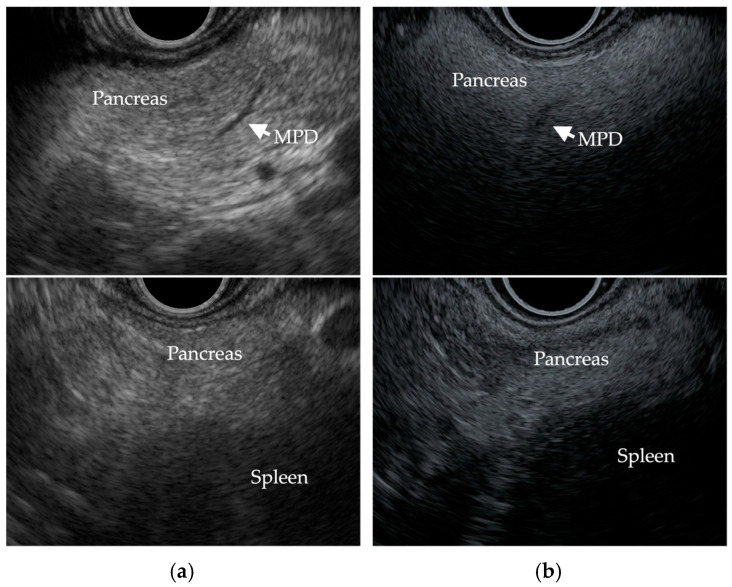
EUS images show (**a**) a non-fatty pancreas with normal echogenicity and a clearly delineated main pancreatic duct (MPD) margin, and (**b**) a fatty pancreas with severe hyperechogenicity, an obscure MPD margin, and increased echogenicity of the pancreas compared to that of the spleen.

**Figure 4 cancers-17-01765-f004:**
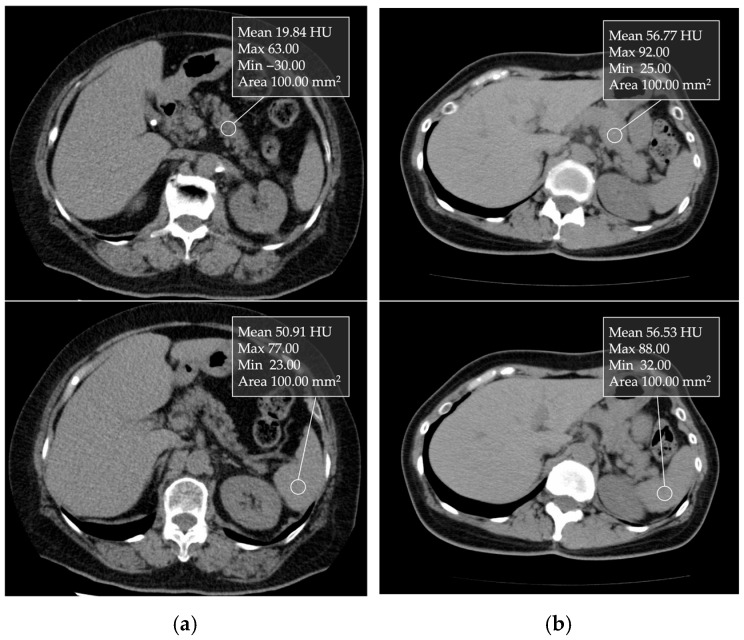
Unenhanced transverse CT images showing 1 cm ROIs placed in the nontumorous pancreatic parenchyma and spleen: (**a**) In a non-fatty pancreas, the pancreatic and splenic CT attenuation values were 56.77 and 56.53 HU, respectively. (**b**) In a fatty pancreas, the values were 19.84 and 50.91 HU, respectively.

**Figure 5 cancers-17-01765-f005:**
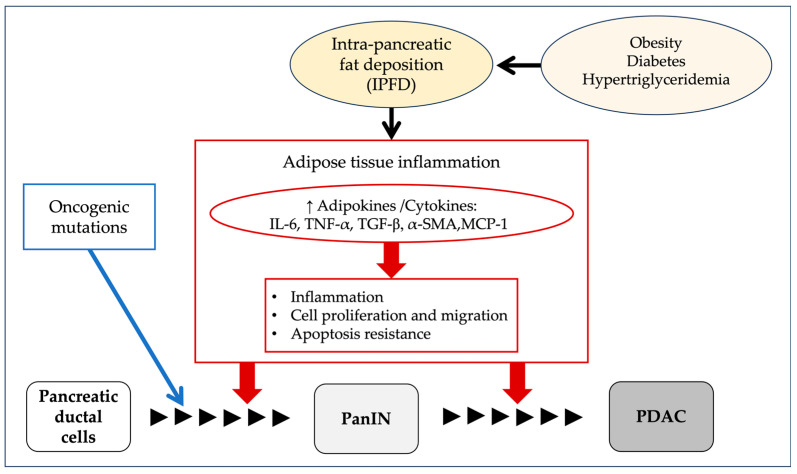
Proposed mechanisms linking intra-pancreatic fat deposition (IPFD) to pancreatic carcinogenesis. IPFD may increase the risk of pancreatic cancer development and progression by promoting the secretion of adipokines that stimulate inflammation, antagonize apoptosis, and enhance cell proliferation and migration. IL-6, including interleukin-6; TNF-α, tumor necrosis factor-α; TGF-β, transforming growth factor-β; α-SMA, α-smooth muscle actin; MCP-1, monocyte chemoattractant protein-1; PanIN, pancreatic intraepithelial neoplasia; PDAC, pancreatic ductal adenocarcinoma.

**Table 1 cancers-17-01765-t001:** Prevalence of fatty pancreas in studies using different imaging modalities.

Study	Year	Modality	No. of Healthy Individuals	Fatty Pancreas *n* (%)
Wu et al. [3]	2013	Ultrasonography	557	72 (12.9%)
Uygun et al. [17]	2014	Ultrasonography	35	5 (14.3%)
Wang et al. [35]	2014	Ultrasonography	8097	1297 (16.0%)
Wong et al. [34]	2014	MRI	685	110 (16.1%)
Lesmana et al. [36]	2015	Ultrasonography	901	315 (35.0%)
Zhou et al. [37]	2016	Ultrasonography	1190	365 (30.7%)

**Table 2 cancers-17-01765-t002:** Effects of various interventions on intra-pancreatic fat deposition (IPFD).

Class of Medication	Study	Year	Total No. of Individuals	Intervention	Effect on IPFD	*p* Value
GLP-1 receptor agonists	Dutour et al. [66]	2016	44	Exenatide	Decreased fat content.	“Non- significant”
	Vanderheiden et al. [67]	2016	71	Liraglutide	Decreased median fat content. Median (IQR) −1.3 (−3.87 to 0.6)	0.056
	Smits et al. [68]	2017	55	Liraglutide	Decreased fat content. −2.4% [95%CI −6.4 to 1.6]	0.24
	Kuchay et al. [69]	2020	88	Dulaglutide	Decreased fat content. −1.4% [95%CI −3.2 to 0.3]	0.106
	Kuriyama et al. [70]	2024	42	Liraglutide	Increased CT attenuation values (pancreas-spleen, HU). Mean ± SD pre 14.3 ± 12.6 post 12.6 ± 10.9	0.0547
DPP-4 inhibitor	Smits et al. [68]	2017	55	Sitagliptin	Decreased fat content. −4.2% [95%CI −8.1 to −0.3]	0.04
SGLT-2 inhibitors	Horii et al. [71]	2021	22	Various SGLT-2 inhibitors (Canagliflozin, Empagliflozin, Dapagliflozin, Ipragliflozin, Luseogliflozin)	Increased CT attenuation values (pancreas-spleen, HU). Median (IQR) pre −20.8 (−34.8 to 14.3) post −14.6 (−29.5 to 7.8)	0.041
	Gaborit et al. [72]	2021	56	Empagliflozin	Described as ‘no significant difference’ without further details.	-
	Hummel et al. [73]	2022	42	Empagliflozin	Decreased mean fat content. Mean ± SD pre 7.1 ± 4.6% post 6.2 ± 4.2%	0.2
	Ghosh et al. [74]	2022	30	Dapagliflozin	Decreased mean fat content. Mean ± SD pre 7.52 ± 5.84% post 5.99 ± 3.98%	0.0083
	Shi et al. [75]	2023	84	Dapagliflozin	Decreased median fat content. Median (IQR) −1.16 (−1.93 to −0.37)	0.033

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
