# Peer review of "Fatty Pancreas: Its Potential as a Risk Factor for Pancreatic Cancer and Clinical Implications"

_cancers, 2025, doi:10.3390/cancers17111765_

Round 1

Reviewer 1 Report

Comments and Suggestions for Authors

In this review, Otsuka et al. discuss the role of fatty accumulation in and outside the pancreas as a risk factor for pancreatic cancer and its clinical implications.

Fat accumulation and adipose tissue inflammation is becoming increasingly interesting in the context of cancer development, including pancreatic cancer.

While this review is of interest, it lacks a more detailed analysis of the literature on the role of fat tissue and pancreatic cancer, especially on the molecular level. This should also include a descriptive figure to illustrate the most important findings on this topic.

Specific comments:

1) Introduction: Line 43 – The sentence on pancreatic cysts is a very general statement considering that there are simple cysts, but also cystic neoplasias that are seen in the pancreas.

2) Pathogenesis: The title of this subsection does not fit to the text of the section except the last paragraph. The authors should consider re-writing the paragraph (or combining it with the pathophysiology section) and really focus on the pathogenesis or re-name the subsection title. 

Figure 1: The figure legend says staining with hematoxylin-eosin. It this really the case? It looks like an IHC with hematoxylin counter stain.

3) Physiopathology and Risk factors:

- Line 104-106: What is associated with T2DM development? The fat or the hypertrophy and hyperplasia of pancreatic cells? This is not clear from the sentence written.

4) Prevalence and Diagnosis:

- Line 114: prevalence “of” FP

- Line 127: Given the lack of standard diagnostic criteria,….

- Figure 2/3/4: For figure 2, figure legend should include a more detailed description of what one can see on the ultrasound images, also maybe by using arrows/arrowheads etc. A picture of fatty versus “non-fatty” pancreas should be shown in order to see differences. Same is necessary for figure 3 and 4. In figure 3 the abbreviation MPD should be included in the figure legend.

In general, the section on diagnosis should be shortened a bit and should focus on the pancreas.

5) Clinical characteristics: In my opinion, there is no real added value to include p-value, PR, CI etc. as it has not been done for all studies described. This enhances readability of the text.

6) Carcinogenesis: This section should include more details and a summarizing figure as it is one major aspect of the review (at least when reading the title; see also comment at the top). Details regarding p-value etc. should be removed (see comment above).

7) Interventions: Are there any studies on the role of exercise and pancreatic fat? If yes, please include.

8) English grammar and spelling check should be performed to improve the readability of the manuscript.

Comments on the Quality of English Language

English grammar and spelling check should be performed to improve the readability of the manuscript.

Reviewer 2 Report

Comments and Suggestions for Authors

The authors give a detailed review on a very interesting topic, the pancreatic steatosis. They mention some of their own experiences (cit 14) related to fatty infiltration and carcinogenesis, but the paper is mainly dedicated to summarize literature data. They organized the recent knowledge in several chapters, focused on the diagnosis of the pancreatic steatosis in humans and its relation to different pancreatic pathologies, being the pancreatic carcinogenesis one of the most important of them.  While the attention to pancreatic steatosis is growing and the number of publications dedicated to it is exponentially increasing in the last years, its real significance in the clinical practice continues to be uncertain. This fact resembles to the situation experienced more than 30 years ago with the hepatic steatosis, which has transformed in one of the most important causes of the liver cirrhosis and its complications, including hepatocellular carcinoma and transplantation. However, the majority of individuals with hepatic steatosis and normal hepatic function do not have a clinically significant liver disease. The prevalence of pancreatic steatosis seems to be similar or at least close to the liver involvement. As a consequence of the very huge numbers, it probably represents a real danger and an important problem for the health system.

Questions, criticism and suggestions:

  • Line 44: “Along with recent attention on metabolic dysfunction-associated steatotic liver disease” – This attention is not so “recent”. Steatohepatitis is a well known diagnosis, as well as other manifestations of MAFLD. On the contrary, pancreatic steatosis is rarely described by radiologists and only exceptionally used as a clinical diagnosis.
  • Line 88: chronic liver disease is mentioned as one of the causes of pancreatic steatosis. I don't know the etiological role of chronic liver disease in the development of fatty pancreas. If it exists, the authors should cite the source of this affirmation.
  • The authors describe and illustrate two types of fatty infiltration: intra- and extralobular and the different importance of these two types in the carcinogenesis, in the formation of PanIN lesions and their progression to invasive cancer. But: How to distinguish these two types in a non-invasive manner?
  • CTscan is a possibility to estimate pancreatic fat content, while not a quantitative manner. Apart of the pancreas/spleen ratio, pancreas-spleen difference also was used. It should be mentioned also in the text, not only in the figure legend. In addition, the study of Kim et al permits a relatively good semiquantitiative evaluation of fat content, based on their comparison of CT parameters with fat content (see figures in the original publication)
  • Finally: Do the authors suggest measure routinely the pancreatic fat content in the everyday clinical practice? Or only in specific cases? On their opinion, which is the best method? Can they give some practical recommendations on the quantitative and semiquantitive measurement of the pancreatic fat content? Which patients need to be followed with repeated images?

Round 2

Reviewer 1 Report

Comments and Suggestions for Authors

The authors have adequately addressed most of my comments. 

Comments on the Quality of English Language

Although significant improvements have been made during the revision process, quality of English could still be improved. 

Author Response

Comments: Although significant improvements have been made during the revision process, quality of English could still be improved. 

Response: We would like to thank the reviewer for thoughtful comments. We would like to have our paper be edited by the publisher, as suggested by the editor.